# Hsa-miR-375/RASD1 Signaling May Predict Local Control in Early Breast Cancer

**DOI:** 10.3390/genes11121404

**Published:** 2020-11-26

**Authors:** Barbara Zellinger, Ulrich Bodenhofer, Immanuela A. Engländer, Cornelia Kronberger, Peter Strasser, Brane Grambozov, Gerd Fastner, Markus Stana, Roland Reitsamer, Karl Sotlar, Felix Sedlmayer, Franz Zehentmayr

**Affiliations:** 1radART—Institute for Research and Development on Advanced Radiation Technologies, Paracelsus Medical University, Müllner Hauptstrasse 48, 5020 Salzburg, Austria; b.zellinger@salk.at (B.Z.); ila.engl@protonmail.com (I.A.E.); f.sedlmayer@salk.at (F.S.); 2Department of Pathology, Paracelsus Medical University, SALK, Müllner Hauptstrasse 48, 5020 Salzburg, Austria; c.kronberger@salk.at (C.K.); k.sotlar@salk.at (K.S.); 3School of Informatics, Communications and Media, University of Applied Sciences Upper Austria, Softwarepark 11, 4232 Hagenberg, Austria; ulrich.bodenhofer@fh-hagenberg.at; 4Institute for Machine Learning, Campus Science Park 3, Johannes Kepler University, Altenbergerstrasse 69, 4040 Linz, Austria; 5Department of Radiation Oncology, Paracelsus Medical University, SALK, Müllner Hauptstrasse 48, 5020 Salzburg, Austria; b.grambozov@salk.at (B.G.); g.fastner@salk.at (G.F.); m.stana@salk.at (M.S.); 6Department of Laboratory Medicine, Paracelsus Medical University, SALK, Müllner Hauptstrasse 48, 5020 Salzburg, Austria; p.strasser@salk.at; 7Department of Gynecology and Obstetrics, Paracelsus Medical University, SALK, Müllner Hauptstrasse 48, 5020 Salzburg, Austria; r.reitsamer@salk.at

**Keywords:** hsa-miR-375, RASD1, early stage breast cancer, local control, predictive markers

## Abstract

Background: In order to characterize the various subtypes of breast cancer more precisely and improve patients selection for breast conserving therapy (BCT), molecular profiling has gained importance over the past two decades. MicroRNAs, which are small non-coding RNAs, can potentially regulate numerous downstream target molecules and thereby interfere in carcinogenesis and treatment response via multiple pathways. The aim of the current two-phase study was to investigate whether hsa-miR-375-signaling through RASD1 could predict local control (LC) in early breast cancer. Results: The patient and treatment characteristics of 81 individuals were similarly distributed between relapse (*n* = 27) and control groups (*n* = 54). In the pilot phase, the primary tumors of 28 patients were analyzed with microarray technology. Of the more than 70,000 genes on the chip, 104 potential hsa-miR-375 target molecules were found to have a lower expression level in relapse patients compared to controls (*p*-value < 0.2). For RASD1, a hsa-miR-375 binding site was predicted by an in silico search in five mRNA-miRNA databases and mechanistically proven in previous pre-clinical studies. Its expression levels were markedly lower in relapse patients than in controls (*p*-value of 0.058). In a second phase, this finding could be validated in an independent set of 53 patients using ddPCR. Patients with enhanced levels of hsa-miR-375 compared to RASD1 had a higher probability of local relapse than those with the inverse expression pattern of the two markers (log-rank test, *p*-value = 0.069). Conclusion: This two-phase study demonstrates that hsa-miR-375/RASD1 signaling is able to predict local control in early breast cancer patients, which—to our knowledge—is the first clinical report on a miR combined with one of its downstream target proteins predicting LC in breast cancer.

## 1. Introduction

Breast cancer is the most common malignant tumor in women, with 93,300 deaths predicted for 2020 in Europe [1]. Carioli estimated that, if the trend between 1989 and 2020 had continued, 32,500 women more would be prone to die in 2020 [1,2]. The epidemiological numbers reflect advances in all related disciplines including early diagnosis by screening programs as well as improved local and systemic treatment. A study by the Early Breast Cancer Trialist Group (EBCTG) revealed that radical treatment of the primary tumor is a prerequisite for long-term cure [3,4]. The standard of care for these patients is breast conserving therapy (BCT) combined with systemic treatment, which—depending on subtype—achieves local control (LC) rates of 95% or more at five years [5,6,7].

Over the past decade, molecular markers were integrated in the staging system for breast cancer [8], paving the way for new prognosticators and predictors such as microRNAs (miRs). These short non-coding RNAs, whose role in breast cancer was first described by Iorio in 2005 [9], consist of 21–25 nucleotides. Based on a comprehensive in vitro and in silico analysis, Tang et al. concluded that the up-regulation of a specific microRNA, i.e., hsa-miR-375, is essential for tumor progression in early breast cancer [10]. In line with this finding, we could demonstrate in a previous publication that the enhanced expression levels of hsa-miR-375 were associated with a higher probability of local relapse [11].

From a translational as well as a clinical point of view, it is important to describe how miRs interfere with cancer pathways. In theory, one miR can regulate the expression of numerous downstream molecules. Hence, in order to select appropriate target proteins that directly influence known cancer pathways, the current analysis builds—first and foremost—on the preclinical data published by Souza et al., who demonstrated that hsa-miR-375 has a functional binding site on the mRNA of the dexamethasone-induced ras-related protein (RASD1) [12]. Both molecules were shown to constitute a positive feedback loop together with the estrogen receptor α (ER-α) [12], which was substantiated by interventional studies with phytoestrogens [13,14,15]. In addition to this, Gao et al. showed that RASD1 overexpression in glioma cells resulted in reduced activity of the Akt/mTOR pathway [16].

Apart from RASD1 and ER-α, Sec23A was considered as a potential target for the current study based on preclinical data in breast [17] and prostate cancer [18]. Sec23A is involved in protein transport from the endoplasmic reticulum to the Golgi apparatus and seems to exert an inhibitory function in cell proliferation. Szczyrba et al. could not only define a binding site for hsa-miR-375 on the mRNA of Sec23A, but also found the protein to be down-regulated in prostate cancer cell lines and tissue specimens [18], which made it worth being included in the current analysis.

It is noteworthy that—apart from the above mentioned analysis by our group [11]—only the investigation by Zhou et al. focused on the role of miRs in LC prediction of early breast cancer [19], while most other studies used progression free survival (PFS) and/or overall survival (OS) as an endpoint [20,21,22,23,24,25]. As an extension of this work [19], the preclinical analysis by de Souza on hsa-miR-375/RASD1 [12] and our own data [11], the aim of the current two-phase study was to answer the question whether hsa-miR-375-signaling through one of the above mentioned targets, i.e., RASD1, ER-α and Sec23A, could predict LC in early breast cancer.

## 2. Methods

### 2.1. Patients

The present analysis is a continuation of a previous publication on the predictive value of hsa-miR-375 in breast cancer [11]. Of the initial 147 patients selected from the institutional database of more than 5000 patients, 81 individuals diagnosed with early breast cancer remained. Due to the age of the pathologic specimen that precluded proper signal detection in some cases and forensic restrictions in others, the cohort had to be reduced to 28 (14 relapses/14 controls) in the pilot phase and 53 (13 relapses/40 controls) in the validation phase. Controls are patients with early breast cancer who did no experience local relapse. Like the previous analysis, this is also a matched pair analysis with the same matching criteria [11]: year of diagnosis, surgery (mastectomy or lumpectomy), radiotherapy (whole breast irradiation with percutaneous or intraoperative boost), age, tumor size, lymph node involvement, grading, histology, hormonal receptor status, her2 status, menopause, and Ki67. In the current report however, due to the reduction in patient numbers for the above-mentioned restrictions, a strict patient-to-patient match was only possible in the pilot phase. The patients of the pilot cohort were not included in the subsequent validation phase.

The endpoint of the current study was LC. This marks a difference between this paper and the majority of publications in the field, which investigated PFS and/or OS. Local relapse was defined as the re-appearance of cancer in the same breast, regardless of whether the former index quadrant was affected or not [26].

All patients gave their informed consent to surgery, radiotherapy and systemic treatment. The study was approved by the local ethics committee in Salzburg (Ethikkommission für das Bundesland Salzburg 415-EP/73/582-2015).

### 2.2. Experimental Design

The current investigation was a two-phase analysis. The first step of the pilot phase was an in silico search for potential targets in five different databases: TargetScan, miRDB, PITA, DIANA, DIANA Cancer (accessed in May 2015). The second step comprised a screen for potential targets in 28 patients (14 with local relapse and 14 matched controls) by means of microarray technology (Affymetrix GeneChip Human Transcriptome Array 2.0^®^). The combined results formed the basis for the validation phase in an independent set of 53 patients performed with droplet digital PCR (ddPCR). The clinical data acquisition and the molecular analyses were performed at the Departments of Radiation Oncology and Pathology at the Paracelsus Medical University Clinics Salzburg, respectively. Blinded to all outcomes, data processing including quality check (Figure A3 and Figure A4), and biostatistics were carried out externally by the Institute for Machine Learning, Johannes Kepler University, Linz. Figure A2 summarizes the experimental design.

### 2.3. Tissue Samples

The current analyses were carried out in the same samples that were used for a previous study published by our group [11]. Immediately after surgery, they were formalin fixed paraffin embedded (FFPE) and archived in the tissue bank of the Department of Pathology without any additional processing. It is noteworthy that the samples were not microdissected to enhance the relative proportion of tumor tissue. Similar to the previous investigation, whole tissue sections were selected, which resulted in a tumor content range of 10% to 90% (see Table 1).

### 2.4. Pilot Phase: Microarray

High through-put RNA expression analysis was performed with the Affymetrix GeneChip Human Transcriptome Array 2.0^®^, which contains a panel of 70,523 RNA transcripts. Five nanogram total RNA from each individual sample was processed with the Affymetrix GeneChip WT Pico Reagent Kit^®^ to obtain 5,5 µg ss-cDNA, which was fragmented, labelled and hybridized to the GeneChip^®^ according to the manufacturer’s instructions. The technical equipment for chip processing was Gene Chip Hybridization Oven 645, Gene Chip Scanner and Gene Chip Fluidics Station 450 Dx (Affymetrix^®^, Thermo Fisher Scientific^®^, Waltham, MA, USA). Microarray data were processed using the RMA method [27,28] from the *oligo* package [29].

### 2.5. Validation Phase: Droplet Digital PCR

#### 2.5.1. Tissue Processing

After routine processing of the tumor specimen, total RNA was isolated from tissue sections with the Maxwell^®^ RSC RNA FFPE Kit (Promega^®^, Madison, WI, USA) according to the manufacturer’s instructions. RNA was eluted in water and stored at −80 °C until further use. RNA was quantified with the QuantiFluor RNA System on Quantus^®^ Fluorimeter (Promega^®^) following the “High Standard Calibration” protocol.

#### 2.5.2. Droplet Digital PCR

Reverse transcription was performed with the iScript^®^ Advanced cDNA Synthesis Kit for RT-qPCR (Bio-Rad^®^, Hercules, CA, USA). A total of 420 ng of each RNA sample was transcribed into cDNA in 20 µL reactions by incubation at 46 °C for 20 min, followed by incubation at 95 °C for 1 min. Then, 1 µL of the cDNA was directly used for gene expression analyses by QX200^®^ ddPCR^®^ EvaGreen system (Bio-Rad^®^) including the following assays: EEF2 (assay ID: dHsaEG5017938) and RASD1 (assay ID: dHsaEG5004980) and self-designed intron-spanning primers for the amplification of RASD1 (F: TCTCCATCCTCACAGGAGAC; R: GTTCTTGAGGCAAGACTTGG). Each 20 µL reaction contained 10 µL 2 × QX200 ddPCR EvaGreen supermix, 1 µL 20 × primers, 1 µL cDNA template and 8 µL RNase/DNase free water. Each sample was prepared in technical duplicates. The cycling conditions were as follows: 95 °C, 5 min; 40 cycles, 96 °C, 30 s; 58 °C, 60 s; 4 °C, 5 min; 90 °C, 5 min; 4 °C, storage; the ramp time was set to 2 °C/s.

#### 2.5.3. Data Analysis

The raw ddPCR data were analyzed with QuantaSoft^®^ (version 1.7.4.0917) supplied by Bio-Rad^®^ in accordance with the MIQE guidelines (minimum information for publication of quantitative digital PCR experiments) [30]. Copies/µL of RASD1 were normalized to the housekeeping gene eucaryotic elongation factor 2 (EEF2) for each sample (https://doi.org/10.1016/j.ymeth.2012.09.012). The normalized values were the basis for further statistical analyses.

### 2.6. Statistics

Potentially prognostic and predictive patient and treatment related parameters were compared between groups by means of the Mann–Whitney U test.

In the pilot phase, de-regulated mRNA transcripts were detected using linear models for microarray analysis (LIMMA) [31]. In accordance with other explorative studies, the *p*-value threshold (α) for the microarray was set at 0.2 [32]. An overly restrictive threshold might have led to losing potential target molecules. To make up for this rather permissive limit for first-order errors (α), the stringency filter to narrow down the vast range of potential target molecules for further validation was the mechanistic proof of a binding site between miRNA and mRNA on the one hand and the result of the in silico search on the other. The statistical data analysis to detect the mRNAs with the highest predictive potential was performed with the potential support vector machine (PSVM), a regularized linear classifier with built-in feature selection [33].

In the validation phase, the levels of hsa-miR-375 and RASD1 were correlated with each other by means of the one-sided Spearman test since an inverse correlation between the two molecules could be assumed based on their biological interaction. LC was estimated with the Kaplan–Meier method. In order to show how far the expression levels of the two molecules are associated with LC, a combined hsa-miR-375/RASD1 marker was generated. Each value was mapped to an index of 1 to 4 representing the quartile which it belonged to. For each sample, the indices were subtracted from each other; while positive results mean higher expression of the microRNA than RASD1, negative values represent the opposite. Zero means “equal” expression levels of hsa-miR-375 and RASD1. The log-rank test was used to compare groups stratified by this combined marker. In order to estimate the predictive value of the combined marker, receiver–operating curve (ROC) analyses were performed for the microRNA/RASD1 ratio. Similar to the microarray, we assumed an α of 0.2 as first-order error, which is the threshold proposed for screening trials [34,35]. Hence, the *p*-value of reference was 0.20.

## 3. Results

### 3.1. Patients

The patient and treatment characteristics of the 81 individuals were similarly distributed between relapse and control groups (Table 1). No significant differences could be detected using the Mann–Whitney U test. The median follow-up in the pilot and validation phase was 130.5 months (range 40–200 months) and 116 months (range 44–214 months), respectively (Table 2). The median time to local relapse was 44 months (range 15–123 months) in the pilot cohort and 77 months (28–140 months) in the validation cohort (Table 2).

### 3.2. Pilot Phase

In order to assess the biological context of hsa-miR-375, potential target genes were identified by two independent approaches: a computational analysis combined with a microarray experiment. Therefore, five different miRNA–target mRNA prediction databases were accessed [36,37,38]. Fifteen targets were predicted both in Target Scan (Release 6.2) and PITA (Venn diagram in Figure A1) including RASD1, which is a validated functional target of hsa-miR-375 in breast cancer cells [12]. The fact that a potential target is listed in at least two databases with different miRNA-mRNA correlation algorithms adds to the reliability of the in silico search results. On top of that, RASD1 was proven in a mechanistic study to have a binding site for hsa-miR-375 [12]. In the very same study, ER-α was shown to be involved in a positive feedback loop between hsa-miR-375 via RASD1 [12]. Sec23A was predicted by DIANA and validated in previous studies in breast [17] and prostate cancer [18].

From the 70,523 probes on the chip, we selected 1433 genes that were predicted to have a binding site with hsa-miR-375 in their 3′UTR using the five databases mentioned in the methods section. The heatmap in Figure 1a summarizes 104 of these potential target molecules that were found to have a lower expression level in relapse patients compared to controls (*p*-value < 0.2). Figure 1b visualizes the proteins whose hsa-miR-375 binding site was predicted by an in silico algorithm and mechanistically proven on a cellular level, i.e., RASD1, Sec23A and ER-α. While RASD1 expression levels were lower in relapse patients than in controls, which reached borderline significance with a *p*-value of 0.058, no difference could be detected for Sec23A (*p*-value = 0.628) and ER-α (*p*-values between 0.546 and 0.902; Figure 2).The microarray data were deposited in the NCBI Gene Expression Omnibus. The microRNA data and the target protein data were deposited under the accession numbers GSE69951 and GSE 156873, respectively; they can be downloaded via the following links:


http://www.ncbi.nlm.nih.gov/geo/query/acc.cgi?acc=GSE69951


http://www.ncbi.nlm.nih.gov/geo/query/acc.cgi?acc=GSE156873.

### 3.3. Validation Phase

As described in the previously mentioned previous publication by our group [11], hsa-miR-375 could significantly predict the probability of local control. In the current validation cohort of 53 patients, the same effect could be observed, although with a higher *p*-value of 0.014 (Figure 3). From the target proteins shown in Figure 2, RASD1 was selected for further validation since the differential expression between relapse, and controls almost reached significance in the microarray (*p*-value = 0.058, Figure 1 and Figure 2). In order to validate this preliminary finding in an independent set of samples, digital droplet PCR (ddPCR), a highly sensitive method to quantify gene expression levels, was used.

This is in coherence with the physiological functioning of a microRNA. Generally, there are two mechanisms at work in the miRNA/mRNA interaction, which both result in decreased mRNA concentrations: a perfect miRNA/mRNA match leads to RISC-mediated dissection of the mRNA molecule, while a less perfect match stops translation with delayed degradation. Hence, changes in the amount of mRNA are an indirect measure for protein concentrations. For lack of suitable RASD1 antibodies, we chose this approach. In the validation cohort, the expression levels of RASD1 and hsa-miR-375 were inversely correlated (one-sided Spearman test, *p*-value = 0.021, Figure A5).

Figure 4 shows the correlation of the combined biomarker hsa-miR-375 and RASD1 to LC. Patients with enhanced levels of hsa-miR-375 compared to RASD1 have a higher probability of local relapse than those with the inverse expression pattern of the two markers. This difference was below the limit of 0.2 for α (log-rank test, *p*-value = 0.069) in the 39/53 (74%) patients who had a clearly higher expression level of one marker (Figure 4a). When the same analysis was performed for the whole cohort including the 14/53 (26%) patients who had an equal expression level of both markers, this difference still persisted (log-rank test, overall *p*-value = 0.177; Figure 4b). The ROC analyses revealed an area under the curve(AUC) of 0.615 (*p*-value = 0.158) and 0.552 (*p*-value = 0.294) for the two cohorts, respectively.

## 4. Discussion

The current two-phase study demonstrates that hsa-miR-375/RASD1 signaling is able to predict local control in early breast cancer. To our knowledge, this is the first clinical report on a miRNA combined with one of its downstream target proteins showing a strong correlation with in-breast tumor control. Since biomarkers have been added to the panel of predictive and prognostic markers in cancer, our work may help to better select patients for BCT.

The first analysis that established the importance of miRs in breast cancer was published 15 years ago by Iorio and co-workers [9]. In this preclinical study, the authors showed that the de-regulated expression levels of 15 miRNAs could discriminate breast cancer from normal tissue. In order to elucidate the functional role of miRs, it is important to know which pathways they interfere in. The most appropriate search process for potential target proteins from the huge number of possible downstream molecules is a miR-mRNA database search combined with the results of mechanistic preclinical studies in cells and rodents. For the current analysis, this approach yielded three potential targets, including RASD1, ER-α and Sec23A, of which only the first was selected for further validation.

Souza et al. provided the mechanistic proof that hsa-miR-375 has a binding site on the UTR-3′-end of the RASD1 mRNA molecule [12]. By gain and loss of function experiments, the authors showed that down-regulated RASD1, which belongs to the Ras superfamily of small G-proteins, resulted in increased cell growth via enhanced ER-α activity [12]. The potential function of RASD1 was investigated in breast and lung cancer cells as well as a xenograft model [39]. High expression levels of this small G-protein stop dysregulated cell growth and may therefore exert anti-cancer activity. Its locus on chromosome 17p11.2 is characterized by frequent loss of heterozygosity and deletions [40], which leads to pro-proliferative signaling in cancer cells [16,39,41,42,43].

The fact that RASD1 is a direct functional target of hsa-miR-375 [12] harbors the potential of a therapeutic approach. This was investigated in two preclinical studies using phytoestrogens [13,14]. Chen showed that the up-regulation of ER-α by the phytoestrogen formononetin and hsa-miR-375 is a universal phenomenon, which is also present in pluripotent endothelial cells (HUVECs) and not only in ER-α positive breast cancer cells [13]. In their study in rodents, Wang et al. demonstrated that calycosin, another phytoestrogen, induces hsa-miR-375 and ER-α, while down regulating RASD1 at the same time [14].

Taken together, the current analysis is the first clinical verification of experimental data on hsa-miR-375 combined with RASD1 in breast cancer. Our results allow us to assume that hsa-miR-375/RASD1 signaling helps to predict local failure, which corroborates the above mentioned preclinical findings [9,12,13,14,15]. The clinical endpoint of the current investigation was LC, while the majority of the studies in the field use PFS and/or OS instead. To date, only one study in a smaller cohort than ours investigated miRs and LC in breast cancer [19], whereas none considered the role of RASD1, let alone the hsa-miR-375/RASD1 combination. Discrepancies with the results of other clinical studies may be related to sample processing. First, in the current analysis, only tumor specimens were compared to each other, while previous studies, e.g., that by Iorio [9], compared tumor and normal tissue leading to a more pronounced differential expression of miRs and their down-stream molecules. Secondly, in the current analysis, tissue sections including tumor stroma were used as opposed to micro-dissected specimen in comparable studies [19,20]. Third, whole tissue sections were analyzed since we believe that a given signaling cascade should retain its discriminative power even if the tumor content is low.

In this context, the question arises how hsa-miR-375/RASD1 interferes in the cancer pathway system. An extensive in silico and in vitro analysis comparing normal versus tumor tissue in more than 3700 cases revealed that up-regulated hsa-miR-375 accelerated cell proliferation by regulating the PI3K/Akt/mTOR pathway [10]. Gao et al. demonstrated the connection between RASD1 and the Akt/mTOR pathway in glioma cells and a xenograft model by means of an array of 18 intracellular signaling molecules [16]. The overexpression of RASD1 reduced the phosphorylation of p-Akt, p-S6 and p-GSK3β, which finally inactivated the Akt/mTOR pathway. Enhanced RASD1 levels led to rearrangement of the cytoskeleton and reduced cell migration [16]. In addition, the clinical data published in the very same study imply that high levels of RASD1 were associated with improved OS [16]. The authors concluded that RASD1 affected cell migration and invasion presumably through Akt-mediated epithelial-mesenchymal transition [16,44], which may translate into clinically detectable differences. In accordance with Gao, the current study substantiates that high levels of RASD1 combined with a low expression of hsa-miR-375 are beneficial in terms of LC. As the report by Gao and our own computational analysis [45] demonstrated, RASD1 interferes with the PI3K/Akt pathway [16] (Figure A7). Therefore, targeting the hsa-miR-375/RASD1 signaling axis could be a rewarding approach for therapeutic studies aimed at inhibiting aberrant cell growth.

To date, clinicaltrials.gov lists a total of 29 studies involving microRNAs in breast cancer, nine of which are completed (accessed 2020/07). These studies cover the following aspects of cancer diagnostics and treatment: response to systemic treatment (10), response to RT (1), response to physical activity and diet (5), including one that focuses on hsa-miR-375, monitoring toxicity (6), and diagnostics (6). As our analysis corroborates preclinical findings, it underlines the usefulness and necessity for inclusion into prospective clinical trials on a larger scale. Thus far, none of these 29 studies on miRs listed in clinicaltrials.gov was conducted with LC as an oncological endpoint.

An obvious weakness of our analysis is the rather permissive threshold for first-order errors (α). However, this is not unusual in explorative studies with the aim of extracting as much as possible of the potentially important information from a huge amount of data [32]. The initial microarray included more than 70,000 target molecules. Nevertheless, RASD1 with a raw *p*-value of 0.058 almost fulfilled the conventional limit of α < 0.05. For reasons described in the methods section, the number of patients had to be reduced to 81 in the current investigation. This may explain why the clinical characteristics such as age, menopause and her2neu were not as evenly distributed as in the previous study [11] (Table 1). Additionally, this rather small number of cases could also be the reason for the moderate predictive power of the combined marker hsa-miR-375/RASD1 in the ROC analyses. While the *p*-value in the subcohort of 39 patients with a clearly inverse correlation between the two markers was still within the limit of 0.2 for first-order errors, in the whole cohort, it was not (Figure A6a,b). Furthermore, the background noise of the multi-level regulatory network, which RASD1 is embedded in, may also play a role. Hence, as the current analysis was designed as a hypothesis-generating study, no definite conclusions can be drawn with respect to sensitivity and specificity.

As for methodological deficiencies, we have to admit that a direct measurement of RASD1 protein concentrations would have certainly strengthened our results. While appropriate antibodies were unavailable, the implementation and validation of a manual staining procedure or the in-house designing of a suitable antibody for RASD1 were beyond the scope of the current project. Using a luciferase reporter construct, de Souza and colleagues provided reliable evidence that RASD1 is a direct target of hsa-miR-375 [12]. These findings were confirmed independently by another group [13]. Similar to our study, neither of these publications included analyses on RASD1 protein levels.

In contrast, these apparent shortcomings are counterbalanced by several strengths. This is the first study that corroborates preclinical data [12] on hsa-miR-375/RASD1 signaling for LC in breast cancer patients. Although the current analysis is retrospective, its two-phase design provides reliable results to constitute a sound basis for prospective investigations. The methodological disadvantage of using whole tissue sections strengthens our results since thereby we keep the specimen processing as close to daily routine as possible, which enhances the transferability of our findings to clinical practice.

## 5. Conclusions

In summary, the current study demonstrates that hsa-miR-375/RASD1 signaling may contribute to predict local control after BCT for early stage breast cancer. Although prospective studies in larger cohorts are warranted, our results enlarge the panel of potential molecular markers for adequate patient selection in the context of tailored therapies.

## Figures and Tables

**Figure 1 genes-11-01404-f001:**
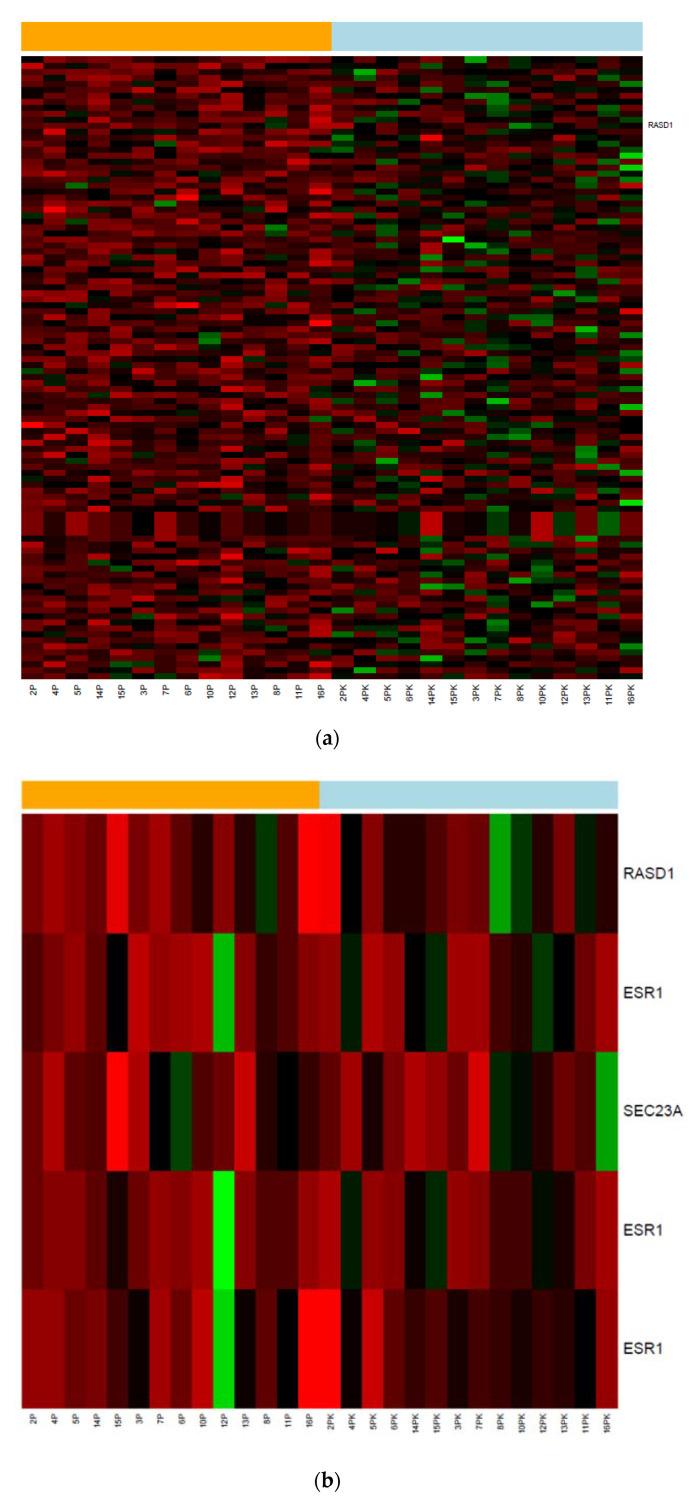
(**a**) The orange bar on top symbolizes the group of patients with local relapse, who were coded with the extension “P” after the number (bottom line), while the blue bar represents the controls (extension “PK”). The microarray (Affymetrix GeneChip Human Transcriptome Array 2.0^®^) contained 70,523 transcripts. According to the in silico analysis, 1433 were known targets of hsa-miR-375, of which 104 were down-regulated in the relapse group (patients beneath the orange bar on top of the heat map) compared to controls (patients beneath the blue bar on top of the heat map): raw *p*-value < 0.20. The transcripts are ordered according to their *p*-values with the lowest on top. RASD1 (highlighted on the right) is number 12 in this list. High and low expressions of a molecule are depicted in green and red, respectively. Abbreviations: P = patients with local relapse, PK = matched control. (**b**) Shows the differential expression of RASD1, ER-α and Sec23A between relapse patients (beneath the orange bar) and controls (beneath the blue bar). High and low expression levels of a molecule are depicted in green and red, respectively. The differential expression of RASD1 reached borderline significance (*p*-value = 0.058), while for the two other targets, the *p*-value was > 0.2. The Affymetrix GeneChip Human Transcriptome Array 2.0^®^ contained three transcripts for ER-α (ESR 1), whereas RASD1 and Sec23A were represented by one transcript each. Abbreviations: P = patients with local relapse, PK = matched control.

**Figure 2 genes-11-01404-f002:**
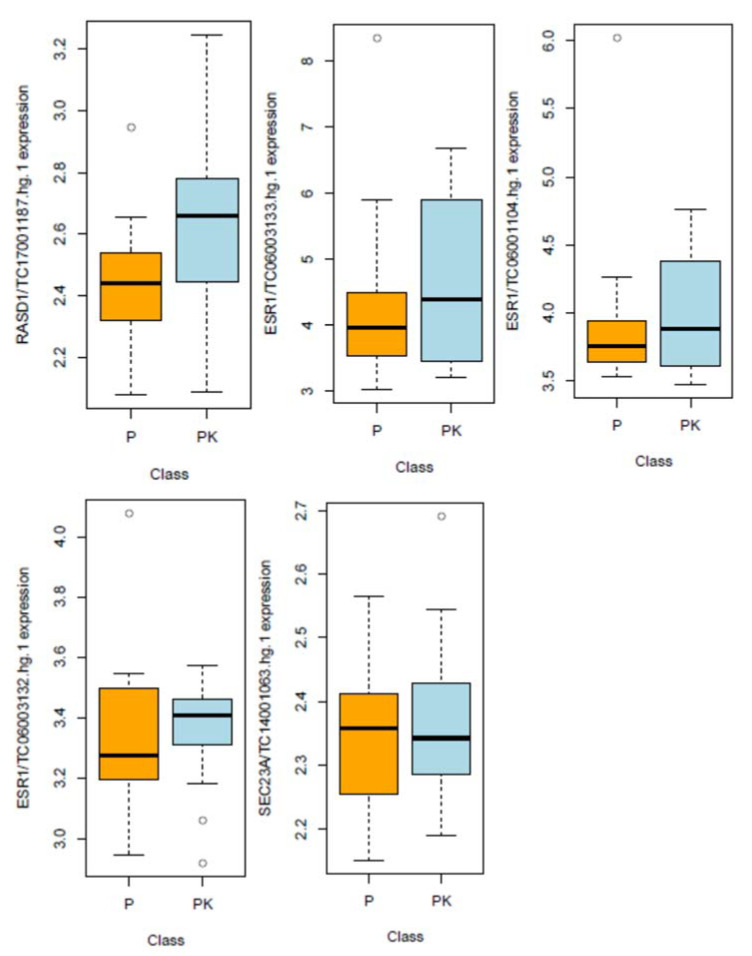
Differential expression between patients and controls for each of the putative target proteins: RASD1, ERα (three transcripts) and Sec23A. The difference in RASD1 expression reached borderline significance (*p*-value = 0.058), while for all other targets, the *p*-value was > 0.2. Abbreviations: P = patients with local relapse, PK = matched control, TC = transcript with the respective number on the Affymetrix GeneChip Human Transcriptome Array 2.0^®^.

**Figure 3 genes-11-01404-f003:**
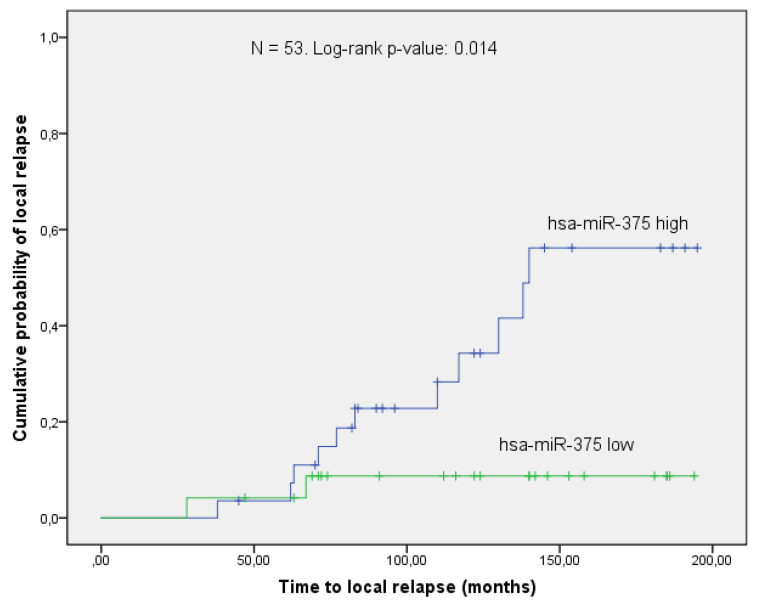
Comparison between patients stratified by the levels of hsa-miR-375 revealed that low concentrations (i.e., below the median) of the micro-RNA are associated with a lower risk of local relapse.

**Figure 4 genes-11-01404-f004:**
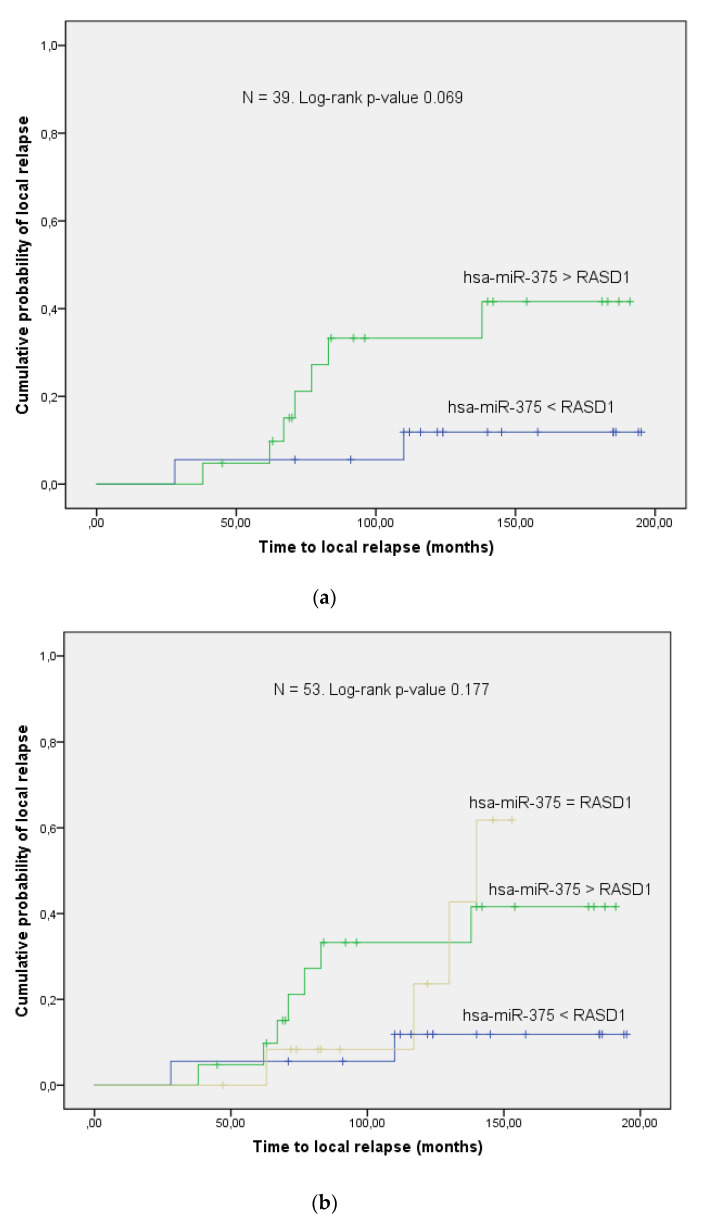
High levels of hsa-miR-375 compared to RASD1 correlate to a higher probability of local relapse. (**a**) The standardized comparison of hsa-miR-375 and RASD1 for each individual patient revealed inverse expression patterns in 39/53 patients. The combined marker hsa-miR-375/RASD1 in these cases was able to predict local control (log-rank *p*-value = 0.069). Hsa-miR-375 > RASD1 means that the expression level of the microRNA is higher than that of the target protein, whereas hsa-miR-375 < RASD1 is the opposite. Hsa-miR-375 = RASD1 signifies that the concentration of both markers is equal. (**b**) As depicted by the clearly separating survival curves, this difference persisted in the whole cohort of 53 patients (log-rank *p*-value = 0.177).

**Table 1 genes-11-01404-t001:** Patient and treatment characteristics.

Patient Characteristics and Treatment			Pilot Phase *n* = 28			Validation Phase *n* = 53		
Parameters			Relapse *n* = 14	Control *n* = 14	*p*-Value	Relapse *n* = 13	Control *n* = 40	*p*-Value
**Patient characteristics**	**Age at diagnosis (years)**	Median	53.5	56	0.84	57	54	0.06
		Range	40–71	41–74		40–79	41–78	
	**Menopause (*n*)**	No	3 (21%)	3 (21%)	0.75	2 (15%)	16 (40%)	0.05
		Yes	7 (50%)	8 (57%)		10 (77%)	18 (45%)	
		Unclear	4 (29%)	3 (22%)		1 (8%)	6 (15%)	
	**T (*n*)**	T1	9 (64%)	8 (57%)	1	11 (85%)	33 (83%)	0.86
		T2	5 (36%)	6 (43%)		2 (15%)	7 (17%)	
	***n* (*n*)**	N0	13 (93%)	12 (86%)	0.77	9 (69%)	30 (75%)	0.55
		N1	1 (7%)	1 (7%)		4 (31%)	10 (25%)	
		N2	0 (0%)	1 (7%)		0 (0%)	0 (0%)	
	**M (*n*)**	M0	14 (100%)	14 (100%)	1	13 (100%)	40 (100%)	1
	**Grading (*n*)**	G1	0 (0%)	0 (0%)	0.87	0 (0%)	4 (10%)	0.56
		G2	7 (50%)	8 (57%)		8 (62%)	22 (55%)	
		G3	7 (50%)	6 (43%)		5 (38%)	14 (35%)	
	**Histology (*n*)**	IDC ^1^	12 (86%)	13 (93%)	0.77	8 (62%)	33 (83%)	0.39
		ILC ^2^	2 (14%)	1 (7%)		3 (23%)	2 (5%)	
		Other	0 (0%)	0 (0%)		2 (15%)	5 (13%)	
	**In situ component (*n*)**	Yes	12 (86%)	8 (57%)	0.11	7 (54%)	20 (50%)	0.81
		No	2 (14%)	6 (43%)		6 (46%)	20 (50%)	
	**Receptors (*n*)**	ER positive ^3^	6 (43%)	8 (57%)	0.54	10 (77%)	30 (75%)	0.89
		ER negative	8 (57%)	6 (43%)		3 (23%)	10 (25%)	
		PR positive ^4^	5 (36%)	7 (50%)	0.54	8 (62%)	18 (70%)	0.57
		PR negative	9 (64%)	7 (50%)		5 (38%)	12 (30%)	
	**her2neu (*n*)**	Positive	8 (57%)	2 (14%)	0.06	4 (31%)	21 (53%)	0.56
		Negative	4 (29%)	9 (64%)		9 (69%)	13 (32%)	
		Not assessable	2 (14%)	3 (21%)		0 (0%)	6 (15%)	
	**Proliferation index (*n*)**	ki67 < 20%	9 (64%)	6 (43%)	0.54	4 (31%)	24 (60%)	0.91
		ki67 > 20%	5 (36%)	7 (50%)		7 (54%)	13 (32%)	
		Not assessable	0 (0%)	1 (7%)		2 (15%)	3 (8%)	
**Treatment**	**Boost**	Intraoperative (*n*)	8 (57%)	7 (50%)	0.8	7 (54%)	21 (53%)	0.22
		Percutaneous (*n*)	6 (43%)	7 (50%)		4 (31%)	19 (47%)	
		None	0 (0%)	0 (0%)		2 (15%)	0 (0%)	
		Intraoperative dose (Gy)	10	10		10	10	
		Percutaneous dose (Gy)	12	12		12	12	
	**WBRT ^5^ dose (Gy)**	Median	54	54	1	54	54	0.2
		Range	52.5–61.2	51.0–57.8		51–57.8	50–54	
	**Surgery (*n*)**	BCT ^6^	14 (100%)	14 (100%)	1	13 (100%)	40 (100%)	1
		Mastectomy	0 (0%)	0 (0%)		0 (0%)	0 (0%)	
	**Re-excision (*n*)**	Yes	6 (43%)	3 (21%)	0.54	5 (38%)	18 (45%)	0.69
		No	8 (57%)	11 (79%)		8 (62%)	22 (55%)	
	**Year of surgery (*n*)**	Before 1998	3 (21%)	4 (29%)	1	5 (38%)	15 (38%)	0.434
		Since 1998	11 (79%)	10 (61%)		8 (62%)	25 (62%)	
	**Chemotherapy (*n*)**	Yes	7 (50%)	5 (36%)	0.21	4 (31%)	11 (28%)	0.86
		No	7 (50%)	9 (64%)		9 (69%)	29 (72%)	
	**Antihormonal treatment (*n*)**	Yes	7 (50%)	6 (43%)	1	9 (69%)	22 (55%)	0.46
		No	7 (50%)	8 (57%)		4 (31%)	16 (40%)	
		Unclear	0 (0%)	0 (0%)		0 (0%)	2 (5%)	
	**Tumor burden in biopsy (%)**	Median	70	60	0.4	70	70	0.76
		Range	10–90	10–90		10–90	30–90	

^1^ Invasive ductal carcinoma, ^2^ Invasive lobular carcinoma, ^3^ Estrogen receptor, ^4^ Progesteron receptor, ^5^ Whole breast radiotherapy, ^6^ Breast conserving therapy.

**Table 2 genes-11-01404-t002:** Clinical outcome.

Patient Characteristics and Treatment		Pilot Phase *n* = 28		Validation Phase *n* = 54	
Parameters		Relapse *n* = 14	Control *n* = 14	Relapse *n* = 13	Control *n* = 41
**Time to local failure (months)**	Median	44	x	77	x
	Range	15–123	x	28–140	x
**Time to distant metastasis (months)**	Median	61	x	71	109
	Range	28–94	x	21–78	x
**Follow-up (months)**	Median	121.5	130.5	95	122
	Range	40–186	72–200	44–214	45–194
**Cancer related deaths (*n*)**		3	0	3	1

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
