# Peer review of "Hsa-miR-375/RASD1 Signaling May Predict Local Control in Early Breast Cancer"

_genes, 2020, doi:10.3390/genes11121404_

Round 1
Reviewer 1 Report
The association of hsa-miR-375 expression with breast cancer relapse has been shown. At the same time, statistically significant relationship between RASD1 expression, as well as the combined action of hsa-miR-375 and RASD1, with relapse was not found. In this regard, attention is drawn to the low level of negative correlation between the hsa-miR-375 and RASD1 mRNA expressions, as follows from the data in Supplement, file 5. Along with the sufficiently convincing evidence of the interaction of the studied microRNAs and the gene, this may indicate the predominant influence of hsa-miR -375 on protein synthesis of the RASD1 gene, rather than on the amount of transcribed mRNA. The authors should be advised to obtain data on the amount of RASD1 protein in connection with the expression of hsa-miR-375 and its relationship with relapse.
Author Response
We would like to thank reviewer 1 for his/her in depth analysis, which has certainly improved the manuscript. Please find our answers to the reviewers’ comments (in italics) below. The text in blue was inserted in the manuscript to account for the requested modifications.
At the same time, statistically significant relationship (…) as well as the combined action of hsa-miR-375 and RASD1, with relapse was not found. In this regard, attention is drawn to the low level of negative correlation between the hsa-miR-375 and RASD1mRNA expressions, as follows from the data in supplement file 5.
Since this is an explorative study, the level of first order error was set at p = 0.2, which is not uncommon in analyses like this (see references 34 and 35). Additional file 5 shows a Spearman correlation between hsa-miR-375 and RASD1 with a one-sided p-value of 0.021, which is markedly below the threshold of 0.2 for first order errors. The use of a one-sided Spearman test is justified in this case since the correlation between the miRNA and the mRNA is supposed to be inverse. But even a two-sided Spearman test would yield a p-value of 0.042, which is below the threshold of 0.2 and even below the 0.05, which is conventionally regarded as significant. We clarified this at the end of the statistics section as follows (lines 190 -191):
Similar to the microarray, we assumed an α of 0.2 as first order error, which is the threshold proposed for screening trials [34,35] Hence the p-value of reference was 0.20.
Along with the sufficiently convincing evidence of the interaction of the studied microRNAs and the gene, this may indicate the predominant influence of hsa-miR-375 on protein synthesis of the RASD1 gene, rather than on the amount of transcribed mRNA. The authors should be advised to obtain data on the amount of RASD1 protein in connection with the expression of hsa-miR-375 and its relationship with relapse.
The issue concerning the expression levels of the protein instead of the mRNA is indeed very important. The current study correlates the microRNA with RASD1-mRNA based on the preclinical work published by de Souza (reference no. 12). Based on this mechanistic approach we showed that hsa-miR-375 and RASD1-mRNA are inversely correlated (additional figure 5). This is in coherence with the well described physiological functioning of a microRNA. Generally, there are two mechanisms at work in the interaction of miRNA and mRNA, which both result in a decrease of the mRNA concentration: a perfect miRNA/mRNA-match leads to the degradation of the mRNA molecule, while a less perfect match stops translation with delayed degradation. A decrease in the amount of the mRNA entails a lower protein concentration. Hence the level of mRNA expression is an indirect measure for the protein concentration.
Using a luciferase reporter construct de Souza and colleagues provided reliable evidence that RASD1 is a direct target of hsa-miR-375: “The observation that modulation of miR-375 caused consistent expression changes in the RASD1-luciferase construct in both cell lines strongly suggests that RASD1 is a functional target of miR-375. Because of the lack of appropriate antibodies, the effect of miR-375 on RASD1 protein levels could not be evaluated.” (de Souza, reference no. 12, p. 9181). These findings are confirmed in a second independent study (Chen, reference no. 13). However, neither of these publications included analyses on RASD1 protein levels. A possible reason for this could be the robustness of the data showing hsa-miR-375 dependent RASD1-mRNA downregulation, which render further analyses, i.e. the direct measurement of protein concentrations, unnecessary. Another – admittedly more likely – reason could be the simple lack of an appropriate antibody, as stated by de Souza. Unfortunately, we also failed to identify a commercial antibody compatible with our immunohistochemistry equipment. Although directly measured data on protein concentration changes would certainly have strengthened our results, the implementation and validation of a manual staining procedure or the in-house designing of a suitable antibody for RASD1 was beyond the scope of the current project. Hence, we are very grateful to reviewer 1 for the valuable suggestion, which we will include in the follow-up project.
Having said this, we have to admit that in some parts of the manuscript we were not very precise with respect to the effect of hsa-miR-375 on RASD1. Accordingly, we modified the results and discussion sections as follows in order to clarify potential misunderstandings:
Lines 262 – 267, Results:
This is in coherence with the physiological functioning of a microRNA. Generally, there are two mechanisms at work in the miRNA/mRNA interaction, which both result in decreased mRNA concentrations: a perfect miRNA/mRNA-match leads to RISC-mediated dissection of the mRNA molecule, while a less perfect match stops translation with delayed degradation. Hence, changes in the amount of mRNA are an indirect measure for protein concentrations. For lack of suitable RASD1 antibodies we chose this approach.
Lines 366 – 373, Discussion:
As for methodological deficiencies, we have to admit that a direct measurement of RASD1 protein concentrations would have certainly strengthened our results. While appropriate antibodies were unavailable, the implementation and validation of a manual staining procedure or the in-house designing of a suitable antibody for RASD1 were beyond the scope of the current project. Using a luciferase reporter construct de Souza and colleagues provided reliable evidence that RASD1 is a direct target of hsa-miR-375 [12]. These findings were confirmed independently by another group [13]. Similar to our study, neither of these publications included analyses on RASD1 protein levels.
In contrast, these apparent shortcomings are counterbalanced by several strengths.

Reviewer 2 Report
The paper by Zellinger et al., is an article suggesting a correlation between has-miR-375 and RASD1 in a pathway to predict local relapse in early breast cancer. Particularly the Authors analyzed an inverse correlation between miR-375 and its downstream target gene RASD1 from 81 early breast cancer cases. I think the paper is really interesting and useful not only for researchers in the field of microRNAs but also for its possible clinical applications. Therefore, minor concerns should be addressed by the Authors:
MINOR CONCERNS:
- In line 89 of the “Introduction” paragraph, the Authors should provide some references about their statement “most other studies used progression free survival (PFS) and / or overall survival (OS) as an endpoint”;
- In “Patients” paragraph, the Authors should clarify who are the controls (are they healthy or are they patients with early breast cancer but without relapses?);
- In “Patients” paragraph, the Authors should explain if the cohort of the pilot phase is included even in the validation phase or if they are two independent cohorts;
- As a consequence of a lack of control group definition, in Table 2 it is not clear why the median time to distant metastasis is 109 months for the controls in the validation cohort;
- the Authors should specify which is the p-value of reference (0.05?);
- For a better understanding of the Figure 1a, the Authors should introduce a legend which explain the two colors used (orange and blue), but also they should upload an image with higher magnification;
- In the caption of the Figure 4, the Authors should specify the meaning of the symbols <, > and =: do they refer to the gene expression?;
- In line 308 of the “Discussion” paragraph, the reference for Wang et al. is missed;
- In the additional file 6, the presentation would be clearer if hsa-miR-375 was put on the top and ERα on the bottom.
Final decision: minor revisions.
Author Response
We would like to thank reviewer 2 for his/her time dedicated to the manuscript. Please find our point-by-point commentaries and the changes in the manuscript based on the reviewer’s helpful suggestions below: reviewer remarks in italics, changes in the manuscript are highlighted in blue.
- In line 89 of the “Introduction” paragraph, the Authors should provide some references about their statement “most other studies used progression free survival (PFS) and / or overall survival (OS) as an endpoint”;
We inserted the following quotations at the end of the above mentioned sentence: Jonsdottir PloS One 2012, Hoppe European Journal of Cancer 2013, Lyng PloS One 2012, Pérez-Rivas PloS One 2014, Tuomarila PloS One 2014, Svoboda Diagnostic Pathology 2012.
- In “Patients” paragraph, the Authors should clarify who are the controls (are they healthy or are they patients with early breast cancer but without relapses?);
Line 100: We inserted the following sentence in the patients section.
Controls are patients with early breast cancer who did not experience local relapse.
- In “Patients” paragraph, the Authors should explain if the cohort of the pilot phase is included even in the validation phase or if they are two independent cohorts;
Line 106: These are two independent cohorts. We clarified this as follows in the patients section.
The patients of the pilot cohort were not included in the subsequent validation phase.
- As a consequence of a lack of control group definition, in Table 2 it is not clear why the median time to distant metastasis is 109 months for the controls in the validation cohort;
By “distant metastasis” we mean disease progression outside the breast. The patients of the control group defined as having early breast cancer without local relapse (see also point 2 above) may – similar to the local relapse patients – develop distant metastasis.
- the Authors should specify which is the p-value of reference (0.05?);
Line 191: We added the following sentence as the very last sentence of the statistics section (line 189):
Hence the p-value of reference was 0.20.
- For a better understanding of the Figure 1a, the Authors should introduce a legend which explains the two colors used (orange and blue), but also they should upload an image with higher magnification;
Lines 229 – 231: We modified the legend for figure 1a as follows and integrated a magnified image of it:
(a) Figure 1a. The orange bar on top symbolizes the group of patients with local relapse, who were coded with the extension “P“ after their number (bottom line), while the blue bar represents the controls (extension „PK“). The microarray (Affymetrix GeneChip Human Transcriptome Array 2.0®) contained 70523 transcripts. According to the in silico analysis, 1433 were known targets of hsa-miR-375, of which 104 were down-regulated in the relapse group compared to controls: raw p-value < 0.20. The transcripts are ordered according to their p-values with the lowest on top. RASD1 (highlighted on the right) is number 12 in this list. High and low expression of a molecule are depicted in green and red colour, respectively. Abbreviations: P = patients with local relapse, PK = matched control.
- In the caption of the Figure 4, the authors should specify the meaning of the symbols <, > and =: do they refer to the gene expression?;
Lines 286 – 291: Accordingly, we modified the legend to figure 4 as follows:
High levels of hsa-miR-375 compared to RASD1 correlate to a higher probability of local relapse. (a) The standardized comparison of hsa-miR-375 and RASD1 for each individual patient revealed inverse expression patterns in 39/53 patients. The combined marker hsa-miR-375/RASD1 in these cases was able to predict local control (log-rank p-value = 0.069). Hsa-miR-375>RASD1 means that the expression level of the microRNA is higher than that of the target protein, whereas hsa-miR-375<RASD1 is the opposite. Hsa-miR-375=RASD1 signifies that the concentration of both markers is equal. (b) As depicted by the clearly separating survival curves this difference persisted in the whole cohort of 53 patients (log-rank p-value = 0.177).
- In line 308 of the “Discussion” paragraph, the reference for Wang et al. is missed;
Line 320: We inserted the missing reference so that the sentence runs as follows:
In their study in rodents, Wang et al. demonstrated that calycosin, another phytoestrogen, induces hsa-miR-375 and ER-α, while down regulating RASD1 at the same time [14].
- In the additional file 6, the presentation would be clearer if hsa-miR-375 was put on the top and ERα on the bottom.
Line 440: This remark is especially helpful. We changed additional file 6 accordingly and believe that the message is transported more clearly. Please see below.

Round 2
Reviewer 1 Report
"Dear colleagues, the difficulties of additional experiments are clear. At the same time, p = 0.2 is acceptable for experiments on microchips, since they are usually followed by verification, but are not used in the final conclusions. In this regard, we can advise you to deepen the statistical processing. The cutoff you applied may not be optimal. Try using ROC analysis for the microRNA/gene expression ratio. You will get not only the result about the presence of the desired relationship, but also the optimal cutoff, which can be used both in the analysis method used by you and in others. Otherwise, it is impossible to say that "Hsa-miR-375/RAZD1 signaling predicts local control" in the absence of statistical significance."
Author Response
Dr. Franz Zehentmayr
Paracelsus Medical University
Müllner Hauptstrasse 48
A-5020 Salzburg
Special Issue: MicroRNAs Applications in Cancer, Therapeutics and Related Toxicities
Salzburg, 15th of November 2020
Dear Dr. Fuentes-Mattei,
Dear Dr. Guo,
Thank you for your ongoing consideration of our manuscript “Hsa-miR-375/RASD1 signaling may predict local control in early breast cancer“.
We found the comments by reviewer 1 to be very helpful and appreciate the feedback. Based on the additional ROC analyses we slightly modified the title, added the respective changes in the methods as well as the results sections and included an expanded critical discussion. We have addressed the reviewer in a point-by-point manner with our responses below. The manuscript has been updated and changes are marked in blue (first round) and green (second round).
We believe that the updated paper has benefited from both review rounds and is much stronger now.
Best regards
Franz Zehentmayr
